# Using Outsourcing Services in Manufacturing Companies

Judyta Kabus [1], Michał Dziadkiewicz [1], Ireneusz Miciuła [2,*] and Marcin Mastalerz [3]

[1] Faculty of Management, Technology University of Częstochowa, 42-200 Częstochowa, Poland; judyta.kabus@pcz.pl (J.K.); michal.dziadkiewicz@wz.pcz.pl (M.D.)

[2] Department of Sustainable Finance and Capital Markets, Faculty of Economics and Finance, University of Szczecin, 70-453 Szczecin, Poland

[3] Department of Computer Science in Management, Faculty of Management, University of Szczecin, 70-453 Szczecin, Poland; marcin.mastalerz@usz.edu.pl

* Correspondence: ireneusz.miciula@usz.edu.pl

**Abstract:** Contemporary economic entities function in various types of cooperation systems, which are primarily aimed at creating a competitive advantage and strengthening themselves in order to meet the requirements of competitors. One solution that can make a significant difference to one's market advantage is outsourcing. It is a response of enterprises to the constantly changing conditions of functioning in a turbulent environment and the emerging new directions and concepts in management. It should be stressed that the choice of outsourcing as a strategy means not only to outsource selected work to external entities, but first of all to retain those competencies of the company that cannot be replaced by anyone. This means that a company must retain a certain sphere of the so-called key areas of activity, which in a positive way distinguish it from the competition and allow it to build an effective market advantage. The main objective of the article is to identify the areas of operation that are the most common subject of outsourcing and the determinants that affect the choice of an outsourcing operator in manufacturing companies in Poland. The variety of aspects of the research subject matter, oscillating around the main objective, has made it necessary to formulate the following research hypotheses: Research Hypothesis H1—The basic criteria determining the selection of an outsourcing operator are: price, quality of services provided and reputation; Research Hypothesis H2—The most common subject of outsourcing is finance and security. The survey was conducted in 2020, in the pre-pandemic period, on a sample of N = 120, including owners/managers of manufacturing enterprises. A non-random sample selection was used. The questionnaires were sent to 200 companies, however, only 126 were completed, of which 6 were not completed in full and were therefore rejected. The verification of the hypothesis was carried out using the chi-square test.

**Keywords:** resources of information; exchange of information; risk; decision making; resource management; management; outsourcing; manufacturing companies/enterprises; benefits; services

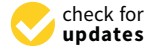

## 1. Introduction

In practice, there is no one universal outsourcing concept that will be optimal for many business entities. The choice of a specific one depends both on internal conditions (e.g., financial resources, employee competence, technologies used, etc.) and external conditions—those from the immediate and further environment of the company. Generally, however, it can be indicated that outsourcing occurs when a specific function (business activity) is purchased from external suppliers [1,2].

The main goal of the outsourcing process is the possibility of focusing the management process on the so-called key competences (i.e., the company's abilities that give it a competitive advantage) [3]. This enables the maximum use of the combination of the company's resources representing its key capabilities in order to build a competitive advantage on the market [4,5]. This is due to the fact that each economic entity is an autonomous unit that

functions in a strictly defined structure, environment and time, which means that an individualised approach is required in the application of outsourcing [6]. The definition of the types of outsourcing is also greatly influenced by multiculturalism (cultural conditions) [7]. They can be interpreted in many different ways, e.g., national cultures in which outsourcing partners operate, cultures of individual companies or individual cultural conditions of individual employees [6,8].

Insourcing is a process opposite to outsourcing, which leads to the inclusion of functions performed by another economic entity into the organizational structure of the enterprise [9,10]. This process is also known as backsourcing [11,12]. As a result, the following types of outsourcing can be distinguished [13]:

- Restorative—used by economic entities threatened with liquidation or in crisis,
- adaptive—used by economic entities that want to develop, be competitive and become market leaders,
- developmental—includes strategic decisions and activities of a developmental or innovative nature; in its essence, it favours the use of the company's market capabilities and opportunities while limiting all threats.

The subject of outsourcing may also include individual functions of the parent economic entity, which are performed within its organizational structure. One of the most concise, and at the same time accurately reflecting the essence of things, definition specifies the functions of the company as: "separate and related logical groups of activities, the continuous performance of which determines the achievement of the company's goals" [14]. Broadly speaking, outsourcing may include the following functions [15]:

- basic;
- auxiliary (complementary);
- managerial.

Due to the fact that the structure of functions is hierarchical and multi-stage, the scope of activities directly related to the allocated function/area may be diversified—very wide, which concerns lower-order functions, or very narrow, which most often includes basic (elementary) functions. The subject of allocation may be, for example, the entire accounting of the enterprise or only a certain partial function—wages [16]. Trocki claims that the set of functions of economic activity creates its functional model, and gives an example of such a model. Figure 1 shows the method of grouping the functions gathered in the functional model of the enterprise created by Trocki [17].

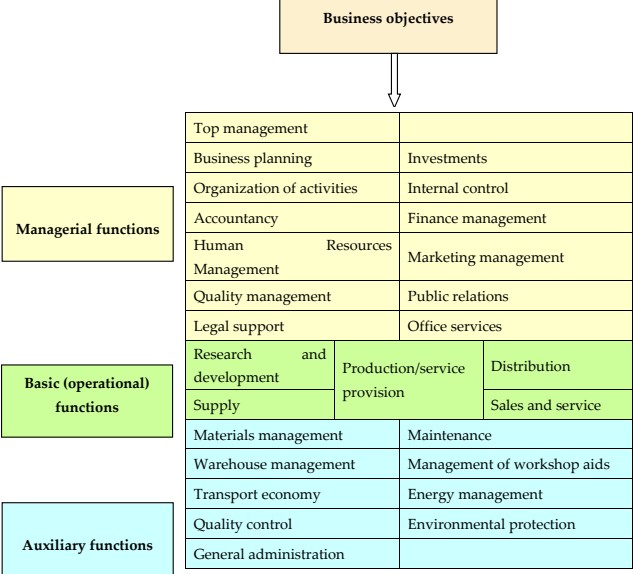

**Figure 1.** Functional model of economic activity.

The model proposed by Trotsky, is an incomplete approach, as contemporary outsourcing should be treated as a method supporting the process of modification or stages of the business model process and shaping the boundaries of the organisation, and not only as activities directly related to the business function. Moreover, the subject of outsourcing does not have to be in all cases the whole process, but can be parts of individual processes, or individual process steps.

Outsourcing can also take a strategic or tactical nature. In the case of the first of the indicated types, the unbundling is permanent and directly related to the development strategy of a given economic entity (client) and should be considered in a long (strategic) time horizon. In the second case, the unbundling is impermanent and does not cover the strategic goals of the economic entity (principal)—it is implemented in shorter periods than the strategic horizon [18–20]. The activity unbundled under outsourcing must undoubtedly be subordinated to the achievement of the adopted economic goals of the enterprise, despite the fact that it means resignation from the influence and subordination [21]. A different division of the types of outsourcing was proposed by Power, Desouza and Bonifazi. These authors distinguished the following criteria for the division of outsourcing processes [22]:

- location;
- depth;
- work.

Bearing in mind the location, internal outsourcing is distinguished (in the same country where the client operates—onshore) [23] as well as external (in the country of the client—offshore) [24]. The following types are distinguished in the offshore outsourcing [25,26]:

- individual—includes individual workstations,
- functional—includes specific functions/fields of activity,
- competence—includes outsourcing activities with simultaneous decision-making by an external entity (contractor).

Referring to the last of the criteria indicated by the authors, i.e., work, there are such types of outsourcing as process and project [27]. In most cases, the process in which the required procedures are very well known to the client, e.g., external service of tax settlements, where cyclicality and efficient document flow are essential. Projects, on the other hand, involve outsourcing non-routine or unique projects that may include, for example, creating personalised software. Managing this type of outsourcing is complicated because the client may not fully understand the complexity of the work. The division of outsourcing was also developed by the Outsourcing Institute in Warsaw [28]. This classification emphasises the services that are provided to enterprises in the outsourcing management process [29]. The types of the processes specified within it are presented in Figure 2 below.

To sum up, it should be noted that outsourcing has specifically defined reasons, which are the goals of outsourcing projects, and to a large extent determines the behaviour of an economic entity [30], which is manifested in the generation of specific events that are important from the perspective of the functioning of each enterprise [31]. In practice, one can distinguish many different divisions of outsourcing, which can be found by analysing the extensive literature on the subject in this field. According to the authors of the study, the essence of the outsourcing process is best reflected by the types based on the model of mutual relations considered as a whole. They include the nature and relationships of these processes (direct and indirect) and their depth, which may be manifested in the mutual benefits of the project, obtained by individual partners involved in it. Therefore, the purpose of this paper is to discuss the factors affecting the selection of an outsourcing service provider and the services most often outsourced to external entities by manufacturing companies in Poland. The article was prepared on the basis of literature studies in the field of management, organisation and implementation of outsourcing projects in enterprises as well as the analysis of surveys in this thematic area. After a critical review of the literature, the authors concluded that outsourcing some activities to external entities is beneficial

to companies, which may be critical to maintaining business in the post COVID-19 pandemic and avoiding bankruptcy. The authors attempted to analyse the phenomenon of outsourcing on the example of Polish manufacturing companies.

**Types of outsourcing**

- simple services - their essence consists in performing basic tasks, the implementation of which may take place in any way; they belong to the group of side tasks, i.e. those that do not directly affect the achievement of the company's goals,
- outtasking services - their essence consists in the implementation/performance of strictly defined and specific tasks that are part of a given process by an external entity, e.g. document digitization, software backups, document printing, telephone services (call centre), data input or recruiting candidates for new employees,
- selective outsourcing services - their essence consists in the provision of services covering complete operational processes of the enterprise, e.g. network services, database management, application hosting, network services, help-desk, calculating employee remuneration, debt collection process, handling incoming correspondence, etc.,
- business process outsourcing services - their essence lies in the implementation of strategic business processes by an external entity, because they affect the enterprise as a whole and are fundamental from the perspective of the business (although they do not constitute the key competences of a given economic entity) and support to a large extent the implementation of its elementary goals, e.g. marketing activities, HR services, fleet management, data centre management, financial and accounting services, loyalty program management, customer service, etc.,
- strategic outsourcing services - they cover specific functions performed in a given economic entity as a whole and consist in a fundamental (revolutionary) change in the way they are performed, with simultaneous unbundling of important processes of this function, which are to be performed by an external entity, e.g. IT or HR outsourcing; one of the types of this kind of outsourcing is transformational outsourcing, the essence of which consists in taking over total responsibility for a given department along with simultaneous restructuring and reconstruction of the processes occurring there.

**Figure 2.** Types of outsourcing according to the Outsourcing Institute.

## 2. The Review of Literature

Outsourcing is now a very dynamic concept of management that covers many different areas of business activity. In the literature on the subject, the areas in which outsourcing is most often used are primarily:

- human resource management [32];
- logistics [5,33];
- production [34];
- IT systems and technologies [35,36].

In the literature on the subject of outsourcing, there are many different classifications of the prerequisites for the use of outsourcing by enterprises. Among the scientists who study the motives of outsourcing some of the activities of enterprises to external entities, one should include in particular Hsiaoa et al. (2010) and Jagdish et al. (2004). The figure below (Figure 3) lists those that were distinguished by Bragg (1998) and the Outsourcing Institute in London [37–39].

**Outsourcing Institute in London**

- improving control and reducing operating costs,
- releasing resources owned by the enterprise for other purposes,
- increasing the company's concentration on its core business areas,
- obtaining resources which the company does not have or resources of better quality,
- risk sharing,
- dealing with functions/areas of activity that are difficult to implement on one's own or are impossi ble to control completely,
- accelerating the emergence of economic benefits that result from restructuring,
- inflow of funds (possibility of obtaining capital).

**S.M. Bragg**

- the need for better management,
- the opportunity to acquire new skills,
- avoiding large investments,
- willingness to focus the attention of the management/managerial staff on the company's strategy,
- increasing flexibility,
- focusing on the core activity of the enterprise, which determines its survival

**Figure 3.** The rationale for outsourcing.

The above approaches are a synthesis of the content identified in the literature on the qualifications and rationale for the use of outsourcing.

It was considered reasonable to cite definitions to base the considerations in this paper on the classification of outsourcing proposed by Bragg and the Institute of Outsourcing in London. Bragg is a significant researcher in the subject of outsourcing, often quoted in works on the subject. His work is regarded as a guide to outsourcing for everyone.

The Institute of Outsourcing in London, in turn, is a well-known provider of out-sourcing services all over the world, familiar with the discussed problem from the practical side.

Outsourcing used by enterprises is one of the key strategies to reduce costs related to production, however, it carries an economic risk that must be taken into account. Particular attention to this issue is paid by Brandes et al. (1997), Chait L.P. (1999), as well as Kremic et al. (2006). The dynamic development of outsourcing took place along with the development of services provided by contractors [40]. Outsourcing in this field of enterprise operation is most often used in the following industries/sectors: automotive, pharmaceutical, clothing, electronics and telecommunications [41,42]. In production enterprises, it

consists in reducing the number of parts/products manufactured in-house and increasing orders from sub-suppliers. The main motive for this action is the aforementioned cost reduction that is possible due to the fact that suppliers provide certain products at lower prices, which results from their specialization and longer production series [43]. Most of the business entities that use outsourcing are being transformed, changing their business model [44,45]. When considering the prerequisites for the use of outsourcing by business entities, one should first of all take the fact that in each enterprise there are many various internal factors into account, the existence of which affects, to a greater or lesser extent, outsourcing decisions. According to Power, Desouza and Bonifazi, at the time of making a strategic assessment, analysis of both strengths and weaknesses, the company should focus on four main elements, among which one can distinguish the following [22]:

- business benefits that include an analysis of the key competences of a given economic entity (they constitute a component of products and/or services produced/offered by a given enterprise and distinguish them from market competitors); supplementary competences are, in turn, necessary to conduct the current activity of the enterprise and have an indirect impact on the products and services it provides;
- operational assessment—essentially consists in diagnosing whether a given enterprise has scenarios that can support outsourcing projects, and whether it has appropriate comparative data and measurement methods that can be used to assess the condition of the economic entity and the competitiveness of the offer presented;
- financial assessment, which should show a direct link with the prerequisites for the use of outsourcing in a given economic entity; in a situation where the basic reason for the use of outsourcing are the costs of fulfilling a function or the costs of implementing selected areas of the company's operations, then it is necessary to conduct a cost analysis;
- risk analysis regarding possible interactions between a given company and an external entity (potential supplier); it is necessary in this case to conduct a risk analysis in the financial, operational and technological context.

Bragg, in his book titled "A Guide to Selecting the Correct Business Unit, Negotiating the Contact, Maintaining Control of the Process", pays particular attention to the outsourcing of accounting functions, logistics services, marketing or cleaning outsourcing. It is these services that he mentions as the most frequently outsourced by enterprises [39]. The subject literature also discusses the problem of outsourcing security services. The authors indicate this service as important for enterprises. Initially, companies ensured the protection of their facilities by using their own employees for this purpose; however, outsourcing security services to specialised companies with appropriate authorisations and trained staff is important from the economic point of view. It reduces personnel costs connected with the necessity personnel costs connected with the necessity to create and maintain own work posts, etc. [46,47]

The approach proposed by the above-mentioned authors makes it possible to precisely define the areas that the company wants to unbundle. The company defines its key competences and focuses solely on them, while other processes are outsourced. Services obtained from the outside significantly support the implementation of strategic goals of a given company, and outsourcing becomes an important element of the strategy [48,49].

## 3. Research Methodology

The use of outsourcing in the world is becoming more and more popular. Many companies decide to use these types of solutions, seeing various benefits in them. However, some business entities decide to independently carry out individual functions and areas of activity, therefore the formulation of the scientific problem in this paper is based on assumptions regarding outsourcing services used in Polish manufacturing companies.

The outsourcing sector was affected by the turmoil caused by the COVID-19 outbreak [50]. However, it can be said that a relatively normal functioning of service providers was observed in this area, even under such exceptional circumstances. Outsourced providers have always presented as one of their strengths the ability to be more detailed

and thoughtful in the field represented. In recent years, we can also observe a reversal of the outsourcing trend leading to a change in the direction of transferring services or investments related to outsourcing services [51,52]. This is related to the prudent approach of companies to the application of outsourcing processes or elements of outsourcing processes. Companies have in mind the desire to achieve benefits for the national economy, including the labour market of the company's home country [53]. Inhibiting the outflow of manufacturing and service processes should, by definition, contribute to, e.g., an increase in employment in the country of origin or shortening of supply chains [54].

A review of the current literature highlights the growing role of outsourcing in business management [55,56]. Outsourcing is studied by scientists in terms of multidimensional aspects, e.g., benefits, threats, applications of IT tools. This topic is also important from the perspective of human resources, cost measurement and the impact of knowledge management on the success of outsourcing. Few literature items discuss/focus on a detailed analysis of the use of outsourcing in Polish manufacturing companies, therefore the primary objective of this article is to analyse the use of outsourcing by these companies. The issues presented in the paper focus on the multifaceted characteristics of outsourcing, with particular emphasis on the company's areas of operation, which are the most frequent subject of outsourcing and the diagnosis of outsourcing service choices in manufacturing companies.

The basis for the formulation of research hypotheses was the study of the literature on the subject and the previously indicated purpose of the study. It was assumed that when choosing outsourcing services Polish companies are guided by price, quality of provided services as well as reputation of the company. The organization to which part of the process is outsourced is responsible for quality to the company for which it performs part of the outsourcing process. The main objective of the study was to identify the areas of operation that are the most common subjects of outsourcing and the determinants that affect the selection of an outsourcing operator in manufacturing companies in Poland. In order to achieve the purpose of the paper, the following research hypotheses were formulated:

**Hypothesis H1.** *The basic criteria determining the selection of an outsourcing operator are: price, quality of services provided and reputation.*

**Hypothesis H2.** *The most common subject of outsourcing is accounting services and security.*

The literature indicates the role of price and quality of services as determinants of the choice of outsourcing services. The most frequently quoted motive for starting the cooperation with a selected outsourcing vendor is the will to reduce operating costs [57,58]. In turn, according to the Laboratory Outsourcing—HRL Technology survey of January 2020, the main motives of enterprises for outsourcing are savings and cost control in the company [59]. Further determinants are the quality of services provided and the company's reputation on the market [60,61]. Based on the literature of Polish researchers, the article identifies accounting and security services as the most common forms of outsourcing. This assumption influenced the formulation of the second hypothesis [62,63]. As a research method, the method of diagnostic survey was used, which consists in collecting data using techniques such as a questionnaire or an interview. The research tool was a structured questionnaire used to verify the hypotheses (H1, H2).

The study was divided into three stages, and these in turn into 10 consecutive phases, including the following stages:

- a preparatory stage consisting of activities related to the determination of the subject, problem, goal and hypotheses of the study, including the development of questions for the questionnaire;
- the stage of the research implementation, including acquiring subjects as well as conducting the study itself;
- the stage of analysis of the obtained research results and their interpretation.
- The survey questionnaire included the following parts:

- introduction, in which there was a request to participate in the study along with the presentation of the purpose of the study plus ensuring anonymity;
- the proper part intended for enterprises using outsourcing services, in which there were six closed questions;
- a certificate with four closed questions.

The answers were given according to a strictly specified scale. The survey was addressed to a representative group of Polish entrepreneurs. The questionnaire was based on the following criteria: The criteria for selecting an outsourcing service provider and the types of outsourcing services included in the questionnaire, developed on the basis of a literature review, provided the basis for verifying the research hypotheses.

The questions included in the questionnaire formed a logical and coherent whole, directly corresponded with the research questions, and the results obtained thanks to them allowed to verify the hypotheses put forward by the authors of the paper. The study was conducted in early 2020, in February and March, in the period prior to the onset of the COVID-19 pandemic. The research was carried out on a sample of N = 120, including owners/managers of production companies from the Silesian Voivodeship. A non-random sample selection was used. The questionnaires were sent to 200 production companies from various areas of Poland; however, only 126 were completed, of which 6 were not completed in full and were therefore rejected. MS Excel was used to process the obtained results, which enabled their graphical and statistical presentation. The verification of the hypotheses was carried out using the chi-square test.

## 4. Discussion

In the literature on the subject, there are also many different concepts of how the outsourcing process should proceed; however, they all share common features. For the purposes of this study, the proposal of Gay et al. (2002), who distinguished seven stages of the outsourcing process, was used, as shown in Figure 4 [37,64,65].

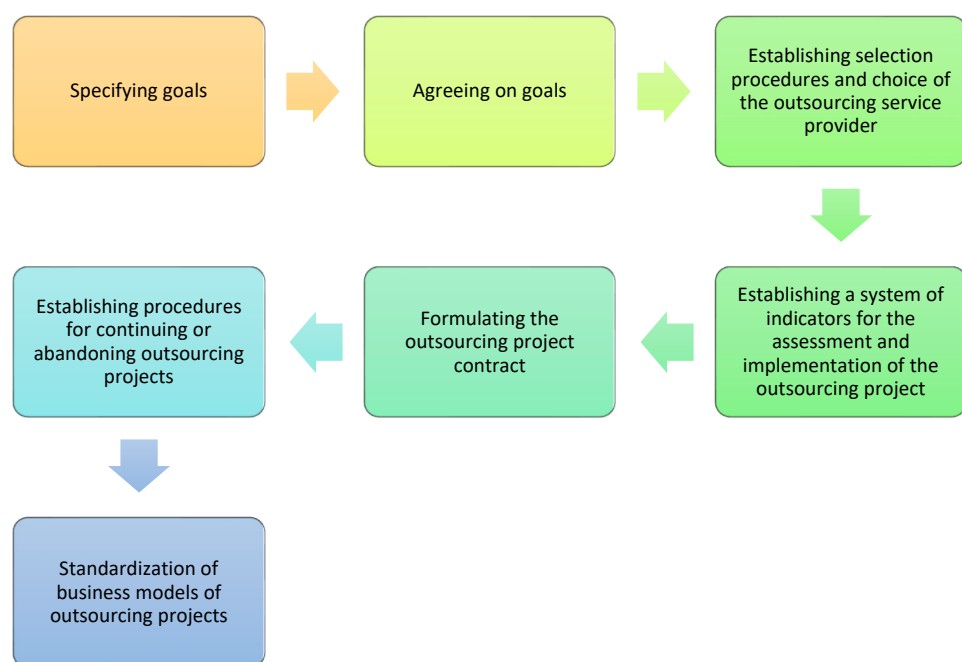

**Figure 4.** Stages of the outsourcing process.

The first stage is to establish your goals. Specifying and identifying the objectives of an outsourcing project is the first and at the same time the most important condition for an enterprise to start outsourcing [66]. In the second stage, activities with an external entity—the service provider—are defined. Agreeing on the purpose with the outsourcing

service provider allows the external entity to evaluate the offer. In practice, it may happen that the supplier has a certain concept, the application of which might ensure the success of the company—the customer, however, it is necessary that they know well and at the same time understand the goals pursued by the service provider [64,67].

An extremely important activity from the perspective of the next stage is to develop selection procedures, including the criteria for selecting the outsourcing service provider. Therefore, it is necessary to precisely establish the terms and conditions of the tender, so as not to block the choice made, e.g., by protests or appeals of other interested parties. In this case, it is important that the company's offer reaches all potential service providers. The company—client should also have a good understanding of potential providers of outsourcing services in the global economy conditions. Making a choice in addition to meeting the conditions for the implementation of a sample contract, the contractors also need to make a comprehensive and in-depth analysis of technological and technical factors and thoroughly examine the economic and financial condition of the service provider, and also define development prospects in a given industry/sector and market [39]. This is extremely important, because if an external entity (i.e., the contractor) is in a bad economic and financial condition or there are indications that they may have significant problems in the future (e.g., a threat of bankruptcy), it generates a number of disruptions, significantly hindering the implementation of the outsourcing project (for example, the transfer of fixed or current assets to the principal may be the subject of a bankruptcy estate in the event of their bankruptcy) [68]. The selection of outsourcing service providers can be made due to many different criteria, such as [22,69,70]:

- geographical criteria (country of the ordering party, country belonging to the European Union, country located on another continent);
- the supplier's scale of operation (small—local, medium—regional/national, large—international, very large—global);
- specialization (process work or performing tasks in the form of projects, implementations);
- experience in providing outsourcing services;
- resources owned by the supplier (production or service capacity, necessary knowledge for the implementation of outsourcing projects).

The next step is to establish a system of evaluation indicators and to implement the outsourcing project. Many information systems used in economic practice have various types of specialised modules that are used in the evaluation of implemented projects (indicators, measures). Enterprises can also easily use them to evaluate the implementation of an outsourcing contract. To make this possible, the system of indicators used should contain certain elements, among which the following can be distinguished [71]:

- selection of adequate indicators (economic, financial and other) so that they reflect the studied phenomenon/area as accurately as possible;
- determining threshold and optimal values of individual indicators;
- determining the frequency of developing individual indicators;
- determining the manner in which individual indicators will be calculated;
- development of an IT system that will be used to calculate the indicators;
- establishing the procedures used to interpret the level (value) of individual indicators; benchmarking can also be used in this case;
- definition of source (primary) information with the methods of obtaining it; it is necessary in order to calculate individual reflecting indicators;
- establishing scenarios of conduct in the case of different levels (values) of indicators (positive and negative).

The next step is to formulate the outsourcing project contract. Each contract should be agreed in a way that will not leave any doubts in the context of its provisions and will minimise the possibility of conflict-generating situations. The first of the essential elements of such a contract should be the determination of potential sanctions (compensations)

for failure to achieve the assumed results. The use of sanctions in the provision of an outsourcing service may have two elementary reasons [72,73]:

- the company–client should receive compensation for the external entity–contractor's failure to meet the specified terms of the contract (e.g., the date and method of delivery or quality) on the delivery of the outsourcing service;
- the penalty for the external entity–outsourcing service provider is aimed at persuading them to pay more attention and care to the proper quality of services in the future.

The contract should contain a provision that any disputes should be resolved amicably. In very difficult and problematic situations, a third party should be used as a conciliator or arbitrator. The last important element of the outsourcing contract should be the definition of detailed conditions for ensuring due diligence in the implementation of the ordered service [74]. This can be achieved by establishing the specific characteristics of a dedicated area of services provided [75]. The sixth stage of implementing the outsourcing process includes establishing procedures for continuing or, in the event of irregularities, abandoning the outsourcing project. It includes the following [76]:

- establishing the criteria on the basis of which the outsourcing agreement with the service provider-contractor will be extended;
- establishing the criteria on the basis of which the outsourcing contract may be terminated.

The issue of responsibility for the quality of the services offered should be emphasised. The definition of the quality expectations of the provided services by the supplier as well as the responsibility for it when the client is not satisfied and complains about the quality of the services should be specified in the outsourcing contract. Responsibility for the quality of the result of the whole process for the customer lies with the organisation that provided part of the outsourcing process. The organisation to which part of the process is outsourced is responsible to the company for which it performs that part of the outsourcing process. This part should also be clearly defined in the contract. Specifying the elements listed above allows to avoid potential conflict situations that might occur between the client and the contractor (e.g., lawsuits). The last part is the standardization of the business model of a given outsourcing project, which should take place both on the part of the company that is the ordering party and the external entity–contractor [77]. It should be noted here that the greatest satisfaction with the implementation of the outsourcing process takes place when it takes a certain amount of time—it is characterised by maturity. Then both sides know each other's needs, which allows for effective cooperation [78]. It also allows for the development of competences in the context of comprehensive management of the outsourcing process and has a positive effect on the quality of this process [63,79]. Summarising the above considerations, it should be noted that outsourcing is not treated by modern enterprises only as a tool enabling the reduction of costs incurred. It is perceived as a concept, the application of which in a company allows for the improvement of efficiency in many areas of activity, e.g., economic, organisational, personal or market. As a result, outsourcing is now a crosscutting concept that affects companies using it in a strategic manner.

## 5. Research Results

A total of 120 business entities carrying out production activities in Poland took part in the study. As shown by the data presented in Figure 5, the survey was dominated by micro-enterprises (employing up to 9 employees), which accounted for 45.8% (55 entities). Small enterprises (employing from 10 to 49 employees) also had a significant share in the total population—30.8% (37 entities) as well as medium-sized enterprises (employing from 50 to 249 employees)—17.5% (21 entities). Large companies (employing 250 and more employees) constituted only 5.8 percent of the total (7 entities).

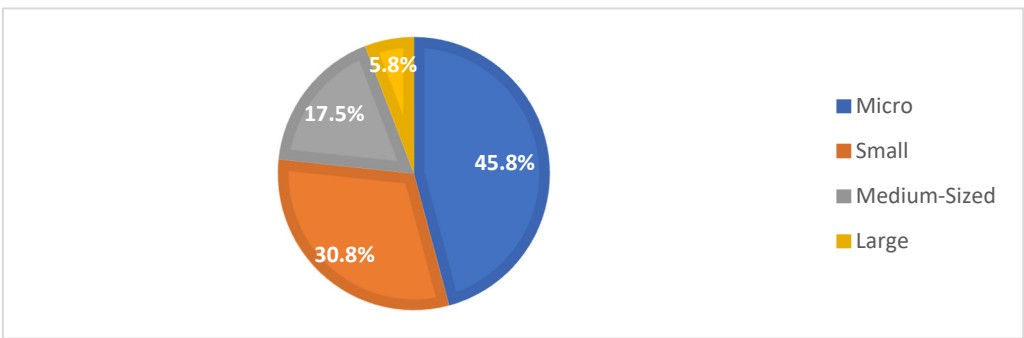

**Figure 5.** Structure of the research sample according to the size of the enterprise.

The business entities participating in the survey conducted activities of various ranges. The largest percentage of them—45.8% (55 entities) operated on a national scale. Enterprises operating on the international market accounted for 39.2 percent (47 entities). The smallest share in the sample was recorded in the case of companies operating locally—5% (6 entities) and regional—10% (12 entities). A detailed summary of this feature of the surveyed enterprises is presented in Figure 6.

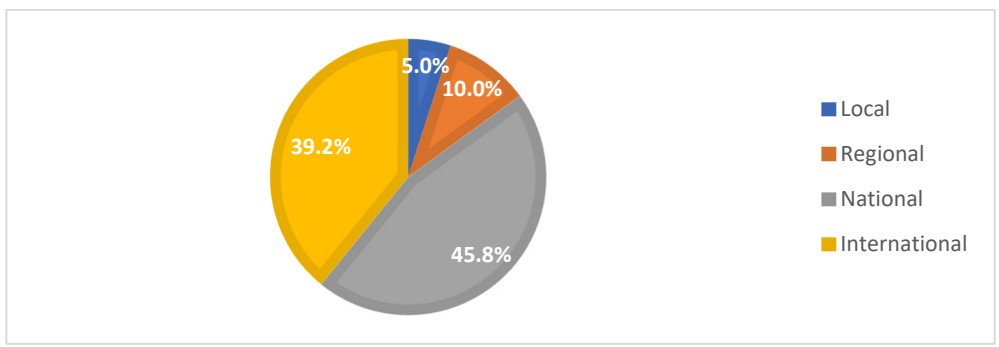

**Figure 6.** Structure of the research sample according to the scope of the conducted activity.

When exploring the data on the length of operation of the surveyed companies on the market, it can be noticed that the oldest companies, operating for over 15 years, constituted 38.3 percent of the total (46 entities) and thus dominated. Enterprises operating on the market for 6 to 10 years accounted for 22.5% (27 entities), and those from 11 to 15 years accounted for 19.2% (23 entities). The total share of enterprises that have been on the market for up to 5 years was 20% (24 entities). Detailed data on the discussed issue is presented in Figure 7.

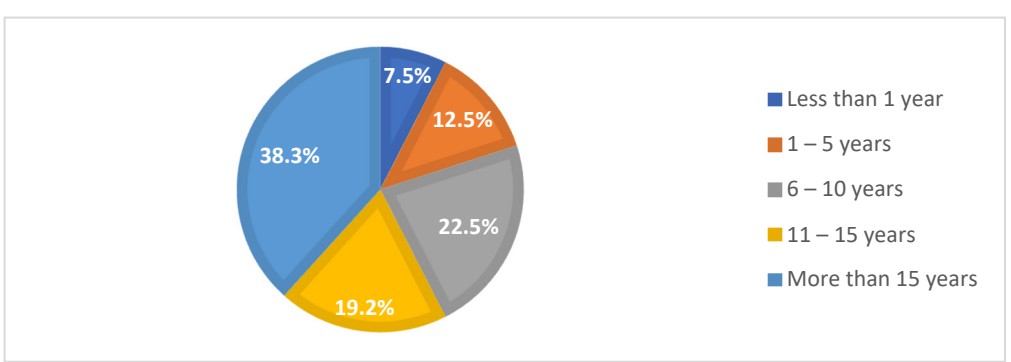

**Figure 7.** Structure of the research sample according to the length of operation on the market. Source: own study.

The last feature characterising the studied enterprises is the location (seat) of the conducted activity (Figure 8).

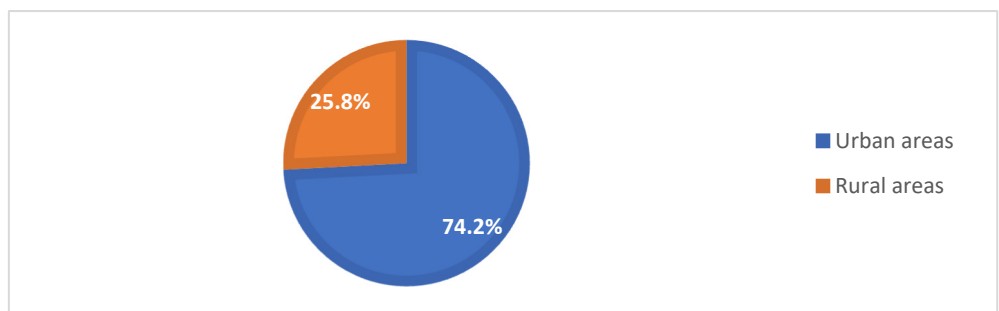

**Figure 8.** Structure of the research sample according to the location of the conducted activity. Source: own study.

According to the presented list, most enterprises are based in urban areas (i.e., cities)—74.2% (89 entities). Enterprises located in rural areas (i.e., villages) constitute only 25.8% (31 entities).

First, it was examined which areas were the subject of outsourcing in the surveyed entities. As can be seen from the information presented in Figure 9, the most frequently outsourced services are:

- security, which is outsourced to external entities by 88.5% of companies using outsourcing (69 entities);
- finances (including bookkeeping), which are outsourced to external entities by 87.2 percent enterprises using outsourcing (68 entities);
- OHS, which is outsourced by 84,6% of companies using outsourcing (66 entities).

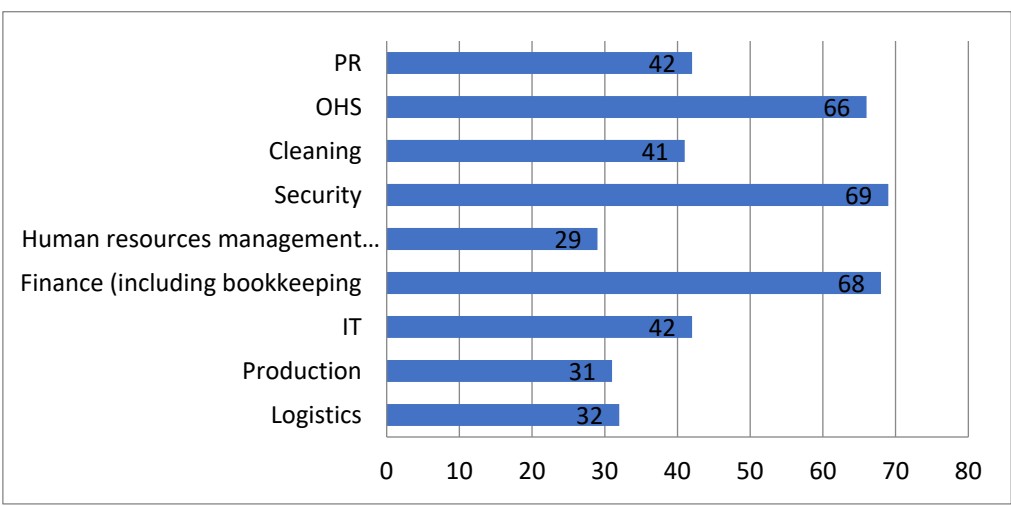

**Figure 9.** Services that are the subject of outsourcing in the studied companies (N = 78). Source: own study.

More than half of the enterprises participating in the study also use the services of external companies, which include public relations, computer network management (IT services) and cleaning. The least frequently unbundled areas of activity are human resources management, production and logistics.

In the further part of the considerations, it is necessary to specify what criteria caused the companies participating in the study to choose a specific outsourcing operator. Figure 10 shows the subjects' answers regarding the importance of the criteria they followed when selecting outsourcing partners. The price was a factor that was of great and very great

significance for 79.5 percent of the subjects (62 entities). Average importance of this factor in the context of the impact on the choice of the outsourcing operator was indicated by 11.5% (9 entities), and little importance by 9% (7 entities). Experience was a criterion of great and very great importance when selecting an outsourcing partner for 73.1% of the subjects (57 entities). A total of 14.1% of the respondents (11 entities) described the share of this criterion as average (i.e., of medium importance, and 12.8% (10 entities) as little).

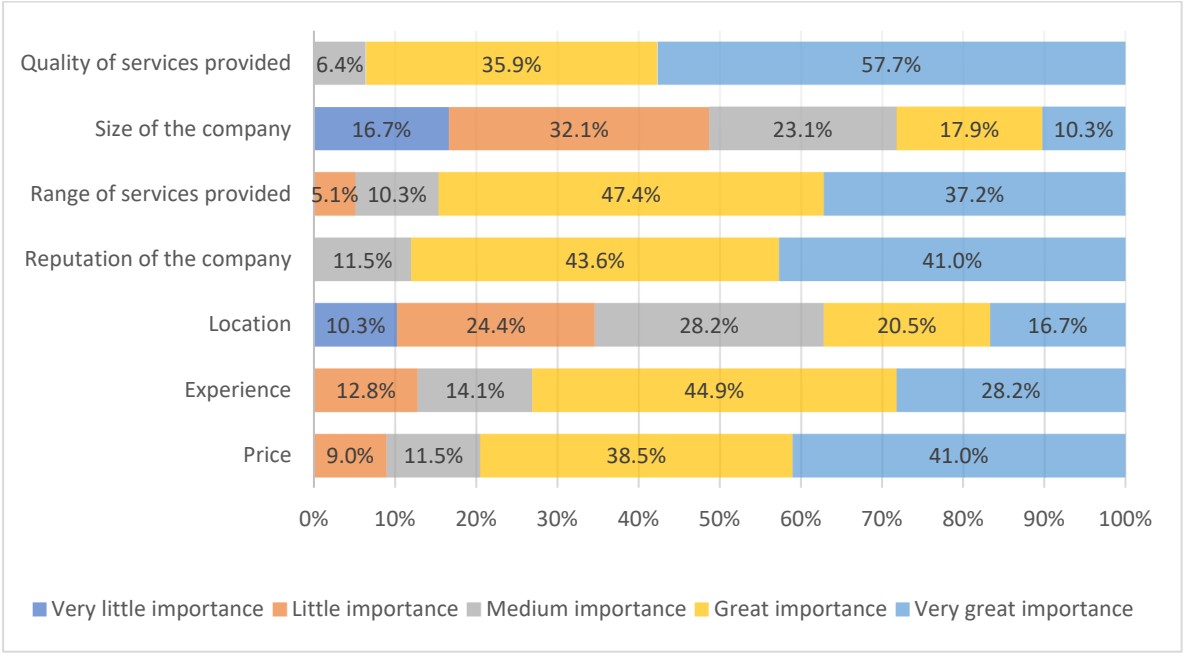

**Figure 10.** Assessment of the impact of selected criteria on the selection of an outsourcing operator (N = 78). Source: own study.

Another factor—location, was an important criterion in choosing an external company for 37.2 percent of enterprises (29 entities). A total of 34.6% (27 entities) did not attach much importance to it, and 28.2 percent (22 entities) considered that it had an average impact on the discussed phenomenon. In the opinion of 84.6% of the subjects (66 entities), the reputation of an outsourcing company was an important criterion that determined the choice. On the other hand, 11.5% (9 entities) believed that this factor was of medium importance when making their choice. The range of services provided by the outsourcing company was of great and very great importance at the time of selection for 84.6% of respondents (66 entities). A total of 10.3% (8 entities) admitted that it affected their decisions to an average extent, and 5.1% (4 entities) described its importance as little. The size of the outsourcing company at the time of commissioning services did not matter much for 48.7 percent of the subjects (38 entities). However, 28.2 percent (22 entities) believed that this factor was significant, and 23.1 percent (18 entities) believed that it had a moderate impact. According to 93.6% of the subjects (73 entities), the quality of the services provided by an outsourcing partner was of great importance when selecting them. The average importance of this criterion was indicated by only 6.4% of the respondents participating in the study (5 entities). In addition to the above considerations, Table 1 presents a synthetic assessment of the impact of selected criteria on the choice of the outsourcing operator. The average was calculated in a similar way to the assessment of satisfaction with outsourcing services. The criteria that had the greatest impact on the selection of the outsourcing operator are marked in green, and the ones that, according to the respondents, had the least impact on the discussed phenomenon, are marked in red.

**Table 1.** Average assessment of the impact of selected criteria on the choice of the outsourcing operator (N = 78).

| Criteria for Selecting an Outsourcing Operator | N | Marginal Response Values | | Average | Mode |
|---|---|---|---|---|---|
| | | Minimum | Maximum | | |
| Price | 78 | 2 | 5 | 4.12 | 5 |
| Experience | 78 | 2 | 5 | 3.88 | 4 |
| Location | 78 | 1 | 5 | 3.09 | 3 |
| Company's reputation | 78 | 3 | 5 | 4.14 | 4 |
| Range of services | 78 | 2 | 5 | 4.17 | 4 |
| Size of the company | 78 | 1 | 5 | 2.73 | 2 |
| Quality of services | 78 | 3 | 5 | 4.51 | 5 |

Source: own study.

When analysing the above list, it can be observed that there are four basic criteria that were used by the subjects in selecting the outsourcing operator, namely:

- quality of services provided (average answer—4.51, mode 5);
- range of services provided (average answer—4.17, mode 4);
- reputation of the outsourcing company (average answer—4.14, mode 4);
- price (average answer—4.12, mode 5).

It is worth emphasising that price is only the fourth factor affecting the decision of the subjects as to the choice of an outsourcing partner. They attach much greater importance to the quality or range of the services provided. However, the factors that have the least impact on the discussed phenomenon are: the location of the outsourcing company (average answer—3.09, mode 3) and its size (average answer—2.73, mode 2). After determining which services are the subject of outsourcing in the surveyed enterprises, the subjects representing them assessed their satisfaction with their quality, which is presented in Figure 11. With regard to companies using outsourcing in the field of logistics, 71.9% of the subjects (23 entities) expressed their satisfaction, 9.4% (3 entities) were dissatisfied, and 18.8% (8 entities) did not care about the quality of services provided. In the case of IT outsourcing, there were no dissatisfied respondents. Positive opinions prevailed among the answers—81% (34 entities). In turn, 19% of the subjects (8 entities) were indifferent to the quality of IT services. Another area, outsourcing of finance (including accounting) was positively assessed by 73.5% of the subjects (50 entities). Dissatisfaction with its quality was expressed by 4.4% of the subjects (3 entities), and for 22.1% (15 entities) it was unimportant. Companies which outsource production to external entities were in most cases satisfied with the quality of these services—74.2% (23 entities), 6.5% (2 entities) expressed their dissatisfaction, and for 19.4% (6 entities) the quality of these services was insignificant. Satisfaction with the outsourcing of human resources management was expressed by 65.5% of the respondents (19 entities). On the contrary, 17.2% of the subjects (5 entities) were dissatisfied. The same number expressed indifference to the quality of outsourcing services in this area.

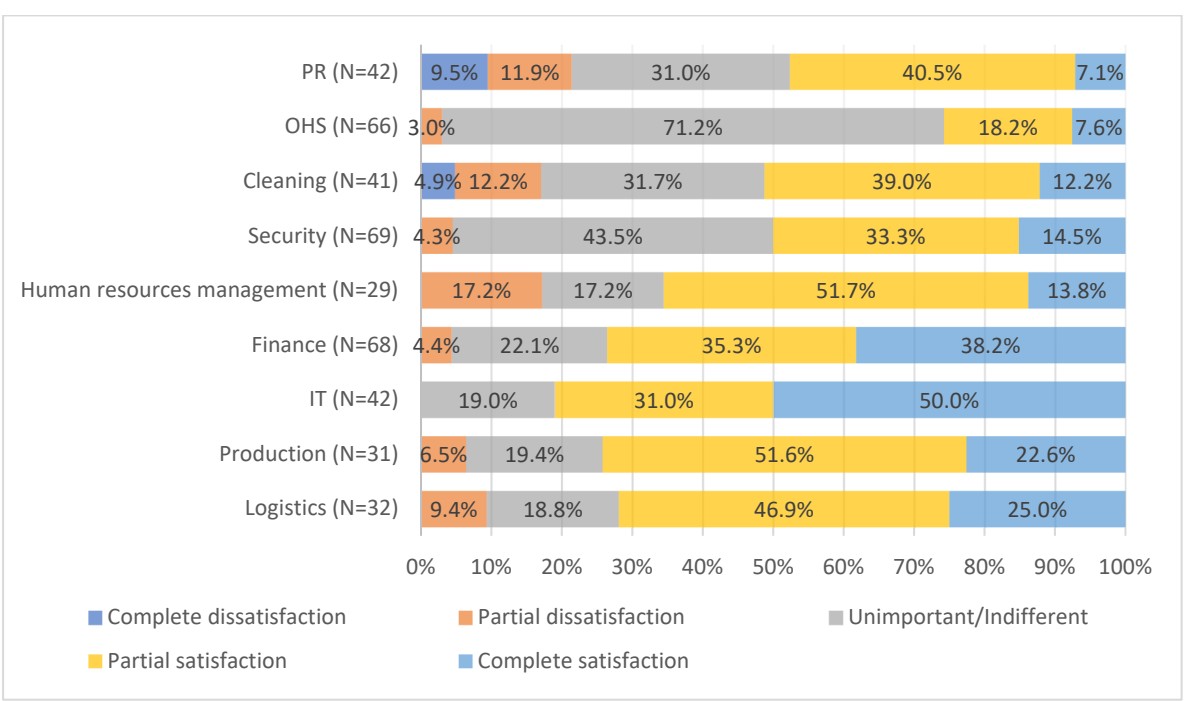

**Figure 11.** Assessment of satisfaction with outsourcing services. Source: own study.

Public relations carried out by external entities was positively assessed by 47.6% of the respondents (20 entities). A total of 21.4% (9 entities) were dissatisfied, and 31% (13 entities) expressed an indifferent opinion. More than half of the subjects using the services of cleaning entities—51.2% (21 entities)—are satisfied with them. A total of 17.1 percent (7 entities) are dissatisfied, and 31.7 percent (13 entities) are indifferent. Security services provided by external entities are unimportant to 43.5% of the subjects (30 entities)—they do not assess them positively or negatively. However, 47.8% (33 entities) are satisfied with them, and 4.3% (3 entities) are dissatisfied. The vast majority of OHS users—71.2% (47 entities) was indifferent to their quality. A total of 25.8% (17 entities) were satisfied, and 3% (2 entities) were dissatisfied. In order to deepen the discussed issue, the assessment of individual outsourcing services is presented in the summary Table 2, where the marginal values of the responses and the weighted average (answer: "complete dissatisfaction"—weighting 1, answer: "complete satisfaction"—weight 5) and the mode are specified. In addition, the services with which the subjects are most satisfied are marked in green, and those with the lowest scores are marked in red.

When exploring the above data, it can be noticed that the subjects participating in the study expressed the greatest satisfaction with the outsourcing services provided in such areas as finance (average answer—4.07, mode 5) and IT (average answer—4.31, mode 5). The respondents were the least satisfied with the outsourcing of OHS (average answer—3.30, mode 3) and public relations (average answer—3.24, mode 4). The analysis of the obtained results made it possible to find answers to the research questions and, as a result, to verify the hypotheses proposed by the authors of the paper. Verification of the H02 hypothesis—the most common subject of outsourcing is finance and security; it required an analysis of the responses provided by company representatives from the perspective of the areas being outsourced in the surveyed entities. The subjects indicated that these include areas such as: security, finance and OHS (i.e., health and safety).

**Table 2.** Average satisfaction rating on outsourcing services (N = 78).

| Area of Outsourcing | N | Marginal Response Values | | Average | Mode |
| --- | --- | --- | --- | --- | --- |
| | | Minimum | Maximum | | |
| Logistics | 32 | 2 | 5 | 3.88 | 4 |
| Production | 31 | 2 | 5 | 3.90 | 4 |
| IT | 42 | 3 | 5 | 4.31 | 5 |
| Finances | 68 | 2 | 5 | 4.07 | 5 |
| HR | 29 | 2 | 5 | 3.62 | 4 |
| Security | 69 | 2 | 5 | 3.45 | 3 |
| Cleaning | 41 | 1 | 5 | 3.41 | 4 |
| OHS | 66 | 2 | 5 | 3.30 | 3 |
| PR | 42 | 1 | 5 | 3.24 | 4 |

Source: own study.

Conducting the chi-square test allowed to examine the relationship between the frequency of using outsourcing services in the field of finance and security, and the frequency of using other outsourcing services. The relationship between the variables was statistically significant. The $\chi^2$ statistic was 163.115 and the *p* value <0.0001 was less than 0.05. Therefore, there is no reason to reject the H02 hypothesis. Thus, it should be recognised that the most common subjects of outsourcing are finance and security. The hypothesis has been confirmed. Verification of the H01 hypothesis (i.e., the basic criteria determining the selection of an outsourcing operator are the price and quality of services provided), was carried out on the basis of the subjects' answers regarding the assessment of the impact of selected criteria on the choice of the outsourcing operator. The respondents decided that the key criteria in this case were: the quality and range of the services provided, the reputation of the outsourcing company and the price. Conducting the chi-square test made it possible to examine the relationship between the assessment of the impact of the price, quality of services provided and the reputation of the outsourcing company, and the assessment of the impact of other criteria on the selection of the outsourcing operator. The relationship between the variables was statistically significant—the value of the $\chi^2$ statistic was 16.4302, and the *p* value was 0.00005 and was lower than the adopted significance level of 0.05. Therefore, there is no reason to reject the H01 hypothesis. Thence, it should be recognised that the basic criteria determining the selection of the outsourcing operator are: the price, the quality of services provided and the reputation of the outsourcing company. The hypothesis has been confirmed.

## 6. Conclusions

The issues presented in this paper allowed not only to analyse the use of outsourcing services in manufacturing companies, but also to identify determinants affecting the choice of a service provider. It should be noted that outsourcing, which is a method rooted in production, currently occupies a strategic place in modern enterprises. As indicated in this paper, the scope of using outsourcing is very wide and may cover virtually every area of the economic entity's activity, except for the core one. The above-mentioned findings are the result of both literature research and empirical research. They confirm the assumed objective. The theoretical considerations and the results of empirical research have shown the accuracy of the research hypotheses and made it possible to achieve the scientific purpose set out in this paper. The presented empirical study shows the following conclusions regarding the verification of the hypotheses:

- the most frequently used outsourcing services are financial and security services;
- the choice of an outsourcing service provider is influenced by the price and quality of the services provided.

Taking a synthetic approach to the article's contribution to the development of knowledge on the issue of outsourcing, it should be noted that its fundamental scientific values are:

- on the theoretical level:
  ○ broadening and systematising the knowledge of outsourcing business activities to external entities;
- on the empirical level:
  ○ identification of key determinants affecting the choice of the outsourcing service provider
  ○ verification of the relationship between the choice of the service provider and the price and quality of services provided
- on the practical level:
  ○ supporting entrepreneurs in conscious and effective planning and implementation of outsourcing services by disseminating research findings (final conclusions).

Summing up, it should be emphasised that outsourcing as a management concept is one of the simplest ways of achieving greater efficiency by enterprises, therefore it is necessary to pay a lot of attention to it. It has become part of the trends set by globalisation processes and the development of information technologies, which enable enterprises to cross country borders and sectors. Therefore, it is necessary to monitor the use of outsourcing by economic entities. The issues related to outsourcing services presented in the work cannot, however, be considered exhaustive, because the discussed problems constitute a multifaceted and interdisciplinary research area. It is recommended to carry out this type of research again in the future. Particular attention should be paid to the impact of the COVID-19 pandemic on the provision of outsourcing services. It is also worth examining the condition of enterprises from the SME sector after the third wave of the pandemic, because they constitute the backbone of the Polish economy, and the use of outsourcing by them may have a positive impact on increasing the effectiveness of their operations and increasing their competitiveness.

**Author Contributions:** Conceptualization, J.K., M.D., M.M. and I.M., methodology, J.K., M.D., M.M. and I.M.; software, J.K., M.D., M.M. and I.M. formal analysis, J.K., M.D., M.M. and I.M.; resources, J.K., M.D., M.M. and I.M. data curation, J.K., M.D., M.M. and I.M.; writing—original draft preparation, J.K., M.D., M.M. and I.M.; writing—review and editing, J.K., M.D., M.M. and I.M.; visualization, J.K., M.D., M.M. and I.M. All authors have read and agreed to the published version of the manuscript.

**Funding:** The project was financed within the framework of the program of the Minister of Science and Higher Education in Poland under the name "Regional Excellence Initiative" in 2019–2022, project number 001/RID/2018/19.

**Institutional Review Board Statement:** Not applicable.

**Informed Consent Statement:** Not applicable.

**Data Availability Statement:** Data are contained within the article.

**Conflicts of Interest:** The authors declare no conflict of interest.

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
