# Peer review of "Using Outsourcing Services in Manufacturing Companies"

_resources, doi:10.3390/resources11030034_

Round 1
Reviewer 1 Report
„The main objective of the study was to identify the areas of operation that are the most common subject of outsourcing and the determinants that affect the selection of an outsourcing operator in manufacturing companies in Poland.“
That objective has been achieved.
„The main goal of the outsourcing process is the possibility of focusing the management process on the so-called key competences (i.e. the company's abilities that give it a competitive advantage).“
This is partly true, but it should be emphasized that choosing outsourcing as a strategy also means choosing another strategy, which is to focus on your own core business.
Figure 1, taken from a relatively old source (2001), discusses managerial, fundamental operational, and complementary functions. This image was taken uncritically, which is not explained. Namely, we need to talk about processes. It is an incomplete figure because processes are cross-functional and broader than functions, and are subject to outsourcing.
The subject of outsourcing does not have to be in all cases the whole process, but can be parts of individual processes, or individual process steps.
The text discusses the importance of quality and mentions in the process step “Formulating the outsourcing project contract,” which is u correct. However, the issue of responsibility for quality should be particularly emphasized. Namely, if part of the process or one of the process steps is outsourced, the responsibility for the quality of the results of the whole process to the customer lies with the organization that provided part of the outsourcing process. The organization to which part of the process is outsourced is responsible to the company for which it performs part of the outsourcing process. And that part should be clearly defined by the contract.
„The use of outsourcing in the world is becoming more and more popular. Many companies decide to use these type of solutions, seeing various benefits in them.“ This is true, however, in recent years there has been a reverse trend, ie that parent companies, for example from Western Europe, return their production from the Far East to which they once outsourced certain processes or parts of them. The reason for this is the change of context, and it wants to achieve some goals of national economies, such as reducing unemployment, due to the shortening of supply chains, for political reasons, etc.
Author Response
Dear Reviewer,
We would like to express our appreciation for the reviews. Thank you very much for suggestions, which were clear and very accurate. We made the necessary corrections. We have incorporated all the suggestions because we agreed with them, and thank you especially for such good suggestions to improve our article.
We would like to refer to the detailed reviewer’s suggestions below:
„The main objective of the study was to identify the areas of operation that are the most common subject of outsourcing and the determinants that affect the selection of an outsourcing operator in manufacturing companies in Poland.“
That objective has been achieved.
Authors’ response: Thank you for the positive reception of the article.
„The main goal of the outsourcing process is the possibility of focusing the management process on the so-called key competences (i.e. the company's abilities that give it a competitive advantage).“ This is partly true, but it should be emphasized that choosing outsourcing as a strategy also means choosing another strategy, which is to focus on your own core business. Figure 1, taken from a relatively old source (2001), discusses managerial, fundamental operational, and complementary functions. This image was taken uncritically, which is not explained. Namely, we need to talk about processes. It is an incomplete figure because processes are cross-functional and broader than functions, and are subject to outsourcing. The subject of outsourcing does not have to be in all cases the whole process, but can be parts of individual processes, or individual process steps.
Authors’ response: We fully agree with the comments. Corrections have been made.
The text discusses the importance of quality and mentions in the process step “Formulating the outsourcing project contract,” which is u correct. However, the issue of responsibility for quality should be particularly emphasized. Namely, if part of the process or one of the process steps is outsourced, the responsibility for the quality of the results of the whole process to the customer lies with the organization that provided part of the outsourcing process. The organization to which part of the process is outsourced is responsible to the company for which it performs part of the outsourcing process. And that part should be clearly defined by the contract.
Authors’ response: According with the opinions, we made the necessary corrections. Relevant paragraphs have been added for a clearer presentation of quantitative data together with an analysis of their implications.
The use of outsourcing in the world is becoming more and more popular. Many companies decide to use these type of solutions, seeing various benefits in them.“ This is true, however, in recent years there has been a reverse trend, ie that parent companies, for example from Western Europe, return their production from the Far East to which they once outsourced certain processes or parts of them. The reason for this is the change of context, and it wants to achieve some goals of national economies, such as reducing unemployment, due to the shortening of supply chains, for political reasons, etc.
Authors’ response: The analysis was extended based on new sources, including those indicated by other reviewers, and the text was redrafted to correspond to the structure of the article. We have also reviewed the suggested bibliographic items and added them in the appropriate places, as well as other current references on the topic in question among scientific journals.
- Malik, A., Lan, J. The role of outsourcing in driving global carbon emissions. Economic Systems Research, 28: 2, 168-18228: 2, 2016, pp. 168-182.
- Radlo, M.J. Offshoring and outsourcing. Implications for the economy and enterprises. Warsaw School of Economics Publishing House, 2013,
- Lai, E., Riezman, R., Wang, P., Outsourcing of innovation. Econ Theory 2009, p. 38
- Szymaniak, A. (Re). Globalization of services: outsourcing, offshoring and shared services centers, WAiP, 2008, Warsaw
- Outsourcing of services in Poland. Report on the survey of companies, 2019. Available online: https://hrl.pl/outsourcing-uslug-w-polsce-raport-z-badania-firm/ (accessed on 04/02/2022).
- Corbett, M.F. Outsourcing Revolution. Why It Makes Sense and How to Do It Right, Dearbon Trade Publishing, 2004, Chicago
- Ren, S., J-F., Ngai E., W., T., Cho V. Examining the determinants of outsourcing partnership quality in Chinese small- and medium-sized enterprises. International Journal of Production Research, Volume 48, 2010 - Issue 2: Modeling and Analysis of Outsourcing Decisions in Global Supply ChainsPages, pp. 453-475, https://doi.org/10.1080/00207540903174965
- Matejun, M. Outsourcing of accounting and tax consultancy Improving management systems in the information society, Volume 2,. StabryÅ‚a A. (ed.), Publishing House of AE in Kraków, 2006 Kraków
- Jarka, S. Status and development prospects of outsourcing in Poland. Scientific Papers of the Warsaw University of Life Sciences, Economics and Organization of Food Economy 93, 2011.
- Skoczylas, P. Outsourcing - as an effective restructuring tool in medical facilities, Entrepreneurship and Management, SAN Publishing House Volume XVII, Issue 4, Part I, 2016, pp. 143–159
- Wojtaszek, H., Miciuła, I. Analysis of factors giving the opportunity for implementation of innovations on the example of manufacturing enterprises in the Silesian province. Sustainability, 2019, 11, 5850, 1-22.
We have incorporated all the suggestions made by the reviewers. Those changes are highlighted within the revised manuscript file with tracked changes. Thanks again for the clear review and suggestions for corrections to improve our article.

Reviewer 2 Report
The concept of the paper is too simple. The paper can be improved with suitable hypotheses, extended variables and rigorous data analysis.
Need to state the tangible outcome of the paper.
Author Response
Dear Reviewer,
We would like to express our appreciation for the reviews. Thank you very much for suggestions, which were clear and very accurate. We made the necessary corrections. We have incorporated all the suggestions because we agreed with them, and thank you especially for such good suggestions to improve our article.
We would like to refer to the detailed reviewer’s suggestions below:
The concept of the paper is too simple. The paper can be improved with suitable hypotheses, extended variables and rigorous data analysis. Need to state the tangible outcome of the paper.
Authors’ response: Thank you for the positive reception of the article. As suggested, the summary has been corrected and completely rebuilt in order to clearly define and present the purpose of the article. Thank you for your attention and pointing to the possibility of improving the structure of the article's goals. We made the necessary corrections. The analysis was extended based on new sources, including those indicated by other reviewers, and the text was redrafted to correspond to the structure of the article. Relevant paragraphs have been added for a clearer presentation of quantitative data together with an analysis of their implications. We have also reviewed the suggested bibliographic items and added them in the appropriate places, as well as other current references on the topic in question among scientific journals.
- Malik, A., Lan, J. The role of outsourcing in driving global carbon emissions. Economic Systems Research, 28: 2, 168-18228: 2, 2016, pp. 168-182.
- Radlo, M.J. Offshoring and outsourcing. Implications for the economy and enterprises. Warsaw School of Economics Publishing House, 2013,
- Lai, E., Riezman, R., Wang, P., Outsourcing of innovation. Econ Theory 2009, p. 38
- Szymaniak, A. (Re). Globalization of services: outsourcing, offshoring and shared services centers, WAiP, 2008, Warsaw
- Outsourcing of services in Poland. Report on the survey of companies, 2019. Available online: https://hrl.pl/outsourcing-uslug-w-polsce-raport-z-badania-firm/ (accessed on 04/02/2022).
- Corbett, M.F. Outsourcing Revolution. Why It Makes Sense and How to Do It Right, Dearbon Trade Publishing, 2004, Chicago
- Ren, S., J-F., Ngai E., W., T., Cho V. Examining the determinants of outsourcing partnership quality in Chinese small- and medium-sized enterprises. International Journal of Production Research, Volume 48, 2010 - Issue 2: Modeling and Analysis of Outsourcing Decisions in Global Supply ChainsPages, pp. 453-475, https://doi.org/10.1080/00207540903174965
- Matejun, M. Outsourcing of accounting and tax consultancy Improving management systems in the information society, Volume 2,. StabryÅ‚a A. (ed.), Publishing House of AE in Kraków, 2006 Kraków
- Jarka, S. Status and development prospects of outsourcing in Poland. Scientific Papers of the Warsaw University of Life Sciences, Economics and Organization of Food Economy 93, 2011.
- Skoczylas, P. Outsourcing - as an effective restructuring tool in medical facilities, Entrepreneurship and Management, SAN Publishing House Volume XVII, Issue 4, Part I, 2016, pp. 143–159
- Wojtaszek, H., Miciuła, I. Analysis of factors giving the opportunity for implementation of innovations on the example of manufacturing enterprises in the Silesian province. Sustainability, 2019, 11, 5850, 1-22.
Authors’ response: Thank you very much for such a useful and developing review and comments supporting the development of the article. We have incorporated all the suggestions made by the reviewers. Those changes are highlighted within the revised manuscript file with tracked changes. Thanks again for the clear review and suggestions for corrections to improve our article.

Reviewer 3 Report
- Row 16-17: „One of the solutions which guarantees an advantage on the market is outsourcing”. There is no guarantee that success will be achieved through the use of outsourcing.
- Row 108 – unclear: „with a structured and standard structure” – what does it mean standard structure or structured structure?
- Row 140: „Sars-Covid-19”? Sars Cov 2 – the virus or Covid-19 pandemic.
- It is not clear why figure 3 uses these sources to present the motives for outsourcing. Needed explanation and justification.
- Row 175-176: It is unclear what is strategic assesment and what is the role of such assesment in the decisions concerning implementation of outsourcing.
- Row 203: „The use of outsourcing in the world is becoming more and more popular” – is it as popular contemporarily, as it was before the beginning of pandemic?
- Rows 208-209: „A review of the current literature highlights the growing role of outsourcing in business management”- there are no examples of sources presented, that reflect such growing role.
- Rows 223-226: what was the basis for the formulation of such research hypotheses? Why price, quality of services provided and reputation are highlighted in hypothesis H01? Why, finance and security are selected as the main areas of outsourcing in hypothesis H02? In addition, finance is a very wide range of activities. What types of financial services are treated as types of service often outsourced?
- Row 255 and further: Assuming that this part of the article discusses the results of the study, it is not clear why the content of e.g. stages of outsourcing is not included in this part.
- Figure 9: It is not clear why these types of services were included in the questionnaire. It should be explained in chapter 3.
- Figure 10: It is not clear what was used as the basis of the choice of such examples of criteria on the selection of an outsourcing operator.
- The authors refer to the Covid-19 pandemic pointing out, that outsourcing is a way to avoid bankruptcy under these conditions. It is worth noting that during the pandemic, it turned out the opposite, that dependence on the service provider can result in disruptions in the business (an example of broken supply chains, especially in offshore outsourcing).
Author Response
Dear Reviewer,
We would like to express our appreciation for the reviews. Thank you very much for suggestions, which were clear and very accurate. We made the necessary corrections. We have incorporated all the suggestions because we agreed with them, and thank you especially for such good suggestions to improve our article.
We would like to refer to the detailed reviewer’s suggestions below:
- Row 16-17: „One of the solutions which guarantees an advantage on the market is outsourcing”. There is no guarantee that success will be achieved through the use of outsourcing.
- Row 108 – unclear: „with a structured and standard structure” – what does it mean standard structure or structured structure?
- Row 140: „Sars-Covid-19”? Sars Cov 2 – the virus or Covid-19 pandemic.
Authors’ response: Thank you for the positive reception of the article. We fully agree with the comments. Corrections have been made.
- It is not clear why figure 3 uses these sources to present the motives for outsourcing. Needed explanation and justification.
- Row 175-176: It is unclear what is strategic assesment and what is the role of such assesment in the decisions concerning implementation of outsourcing.
Authors’ response: As suggested, this sections has been corrected and completely rebuilt in order to clearly define and present the purpose of the article. Thank you for your attention and pointing to the possibility of improving the structure of the article's goals. We made the necessary corrections.
- Row 203: „The use of outsourcing in the world is becoming more and more popular” – is it as popular contemporarily, as it was before the beginning of pandemic?
- Rows 208-209: „A review of the current literature highlights the growing role of outsourcing in business management”- there are no examples of sources presented, that reflect such growing role.
Authors’ response: We made the necessary corrections.
- Rows 223-226: what was the basis for the formulation of such research hypotheses? Why price, quality of services provided and reputation are highlighted in hypothesis H01? Why, finance and security are selected as the main areas of outsourcing in hypothesis H02? In addition, finance is a very wide range of activities. What types of financial services are treated as types of service often outsourced?
- Row 255 and further: Assuming that this part of the article discusses the results of the study, it is not clear why the content of e.g. stages of outsourcing is not included in this part.
Authors’ response: The analysis was extended based on new sources, including those indicated by other reviewers, and the text was redrafted to correspond to the structure of the article. Relevant paragraphs have been added for a clearer presentation of quantitative data together with an analysis of their implications. Thank you very much for the broader presentation of the issues and thus the opportunity to improve my own research. Very valuable support in the development of the article has been revised and applied.
- Figure 9: It is not clear why these types of services were included in the questionnaire. It should be explained in chapter 3.
- Figure 10: It is not clear what was used as the basis of the choice of such examples of criteria on the selection of an outsourcing operator.
Authors’ response: According with the opinions, we made the necessary corrections. Relevant paragraphs have been added for a clearer presentation of quantitative data together with an analysis of their implications.
- The authors refer to the Covid-19 pandemic pointing out, that outsourcing is a way to avoid bankruptcy under these conditions. It is worth noting that during the pandemic, it turned out the opposite, that dependence on the service provider can result in disruptions in the business (an example of broken supply chains, especially in offshore outsourcing).
Authors’ response: Thank you very much for such a useful and developing review and comments supporting the development of the article. We have also reviewed the suggested bibliographic items and added them in the appropriate places, as well as other current references on the topic in question among scientific journals.
- Malik, A., Lan, J. The role of outsourcing in driving global carbon emissions. Economic Systems Research, 28: 2, 168-18228: 2, 2016, pp. 168-182.
- Radlo, M.J. Offshoring and outsourcing. Implications for the economy and enterprises. Warsaw School of Economics Publishing House, 2013,
- Lai, E., Riezman, R., Wang, P., Outsourcing of innovation. Econ Theory 2009, p. 38
- Szymaniak, A. (Re). Globalization of services: outsourcing, offshoring and shared services centers, WAiP, 2008, Warsaw
- Outsourcing of services in Poland. Report on the survey of companies, 2019. Available online: https://hrl.pl/outsourcing-uslug-w-polsce-raport-z-badania-firm/ (accessed on 04/02/2022).
- Corbett, M.F. Outsourcing Revolution. Why It Makes Sense and How to Do It Right, Dearbon Trade Publishing, 2004, Chicago
- Ren, S., J-F., Ngai E., W., T., Cho V. Examining the determinants of outsourcing partnership quality in Chinese small- and medium-sized enterprises. International Journal of Production Research, Volume 48, 2010 - Issue 2: Modeling and Analysis of Outsourcing Decisions in Global Supply ChainsPages, pp. 453-475, https://doi.org/10.1080/00207540903174965
- Matejun, M. Outsourcing of accounting and tax consultancy Improving management systems in the information society, Volume 2,. StabryÅ‚a A. (ed.), Publishing House of AE in Kraków, 2006 Kraków
- Jarka, S. Status and development prospects of outsourcing in Poland. Scientific Papers of the Warsaw University of Life Sciences, Economics and Organization of Food Economy 93, 2011.
- Skoczylas, P. Outsourcing - as an effective restructuring tool in medical facilities, Entrepreneurship and Management, SAN Publishing House Volume XVII, Issue 4, Part I, 2016, pp. 143–159
- Wojtaszek, H., Miciuła, I. Analysis of factors giving the opportunity for implementation of innovations on the example of manufacturing enterprises in the Silesian province. Sustainability, 2019, 11, 5850, 1-22.
We have incorporated all the suggestions made by the reviewers. Those changes are highlighted within the revised manuscript file with tracked changes. Thanks again for the clear review and suggestions for corrections in points to improve our article.

Round 2
Reviewer 2 Report
With the overall improvement, the paper looks much better now.
Author Response
Thank you for the positive reception of the article and thank you especially for such good suggestions to improve our article.
Reviewer 3 Report
Outsourcing Institute cannot be treated as provider of logistics services. It is rather the association developing and popularizing knowledge and experience in the field of outsourcing
The explanation of typology of reasons for outsourcing included in Table 3 is still not sufficient. It is not known why these approaches (of Bragg and Outsourcing Institute) were presented. It is not known whether they are a generalization of the approaches presented in the sources mentioned earlier or they represent the content included in all the sources. There is still no substantive justification.
As to H01 as well as H02: The formulation of each research hypothesis must be preceded (supported) by careful study of sources. Sources have to be presented in the text.
Further:
"making a strategic assessment" means not only an analysis of the key success factors.
Rows 228-239: unambiguous, authoritarian statements without providing any sources (studies, articles, books) on the basis of which they were developed.
Author Response
Thank you for the positive reception of the article. We have incorporated all the suggestions made by the reviewers. We made the necessary corrections. Those changes are highlighted within the revised manuscript file with tracked changes. Thanks again for the clear review and suggestions for corrections to improve our article.